# Association between longitudinal blood pressure and prognosis after treatment of cerebral aneurysm: A nationwide population-based cohort study

Jinkwon Kim[1,2], Jang Hoon Kim[3], Hye Sun Lee[4], Sang Hyun Suh[5]*, Kyung-Yul Lee[1]*

1 Department of Neurology, Yonsei University College of Medicine, Seoul, Korea, 2 Department of Neurology, CHA Bundang Medical Center, Seongnam, Korea, 3 Department of Neurosurgery, Yonsei University College of Medicine, Seoul, Korea, 4 Biostatistics Collaboration Unit, Yonsei University College of Medicine, Seoul, Korea, 5 Department of Radiology, Gangnam Severance Hospital, Yonsei University College of Medicine, Seoul, Korea

☯ These authors contributed equally to this work.
* suhsh11@yuhs.ac (SHS); kylee@yuhs.ac (KYL)

**Data Availability Statement:** The Korea National Health Insurance Service, the data source of current study, only allowed the release of summary data or analytic result of health claims data in order

## Abstract

### Background

High blood pressure is a major risk factor for the development and rupture of cerebral aneurysm. Endovascular coil embolization and surgical clipping are established procedures to treat cerebral aneurysm. However, longitudinal data of blood pressure after the treatment of cerebral aneurysm and its impact on long-term prognosis are not well known.

### Methods

This retrospective cohort study included 1275 patients who underwent endovascular coil embolization (n = 558) or surgical clipping (n = 717) of cerebral aneurysm in 2002–2015 using the nationwide health screening database of Korea. Systolic and diastolic blood pressure of patients were repeatedly obtained from the nationwide health screening program. We performed a multivariate time-dependent Cox regression analysis of the primary composite outcome of stroke, myocardial infarction, and all-cause death.

### Results

During the mean follow-up period of 6.13 ± 3.41 years, 89 patients suffered the primary outcome. Among the total 3546 times of blood pressure measurement, uncontrolled high blood pressure (systolic ≥140 mmHg or diastolic ≥90 mmHg) was 22.9%. There was a significantly increased risk of primary outcome with high systolic (adjusted HR [95% CI] per 10 mmHg, 1.16 [1.01–1.35]) and diastolic (adjusted HR [95% CI] per 10 mmHg, 1.32 [1.06–1.64]) blood pressure.

to prevent misuse of personal information even it is fully anonymized. Requests for access to NHIS data can be made through the homepage of National Health Insurance Sharing Service [http://nhiss.nhis.or.kr/bd/ab/bdaba021eng.do]. To gain access to the data, a completed application form, a research proposal and the applicant's approval document from the institutional review board should be submitted to and reviewed by the inquiry committee of research support in NHIS. The NHIS-NSC data were fully anonymized and did not contain any identifiable information.

**Funding:** This work was supported by the Basic Science Research Program through the National Research Foundation of Korea (http://www.nrf.re.kr/eng/index) funded by the Ministry of Education (NRF-2017R1D1A1B03033382 to JK, NRF-2020R1I1A1A01060447 to JK). The funders had no role in study design, data collection and analysis, decision to publish, or preparation of the manuscript.

**Competing interests:** The authors have declared that no competing interests exist.

## Conclusions

High blood pressure is prevalent even in patients who received treatment for cerebral aneurysm, which is significantly associated with poor outcome. Strict control of high blood pressure may further improve the prognosis of patients with cerebral aneurysm.

## Introduction

Cerebral aneurysm, an abnormal focal dilation or ballooning of the intracranial cerebral artery wall, occurs in about 1–5% of the general population [1]. Although most cases of cerebral aneurysm are asymptomatic, some are susceptible to further dilatation and rupture resulting in subarachnoid hemorrhage (SAH), a life-threatening stroke [2]. Because nontraumatic SAH is mostly caused by the rupture of cerebral aneurysms, the management strategy for cerebral aneurysm to reduce the worldwide health burden of SAH is clinically challenging. Endovascular coil embolization and surgical clipping are two established treatments for ruptured and unruptured cerebral aneurysm [3, 4]. Beyond the interventions, the control of risk factors is also desirable for high-risk patients with cerebral aneurysm [5].

Hypertension is recognized as a major risk factor for the development, enlargement, and rupture of cerebral aneurysm [6]. Hemodynamic stress and inflammation induced by high blood pressure could lead to arterial wall damage and dilation, resulting in the growth and rupture of cerebral aneurysm [1, 7]. Therefore, strict control of blood pressure is considered essential in the management of cerebral aneurysm [5]. However, evidence for the relationship between blood pressure and prognosis in patients with cerebral aneurysm are insufficient, especially in those who underwent coil embolization or surgical clipping. The aims of this study are to 1) obtain the longitudinal data of blood pressure after treatment of cerebral aneurysm in real-world patients; and 2) evaluate the impact of longitudinal blood pressure on long-term prognosis using data from the nationwide health screening database in Korea.

## Materials and methods

### Study design and data source

This retrospective cohort study used data from the National Health Insurance Service-National Health Screening Cohort (NHIS-HEALS) in Korea [8]. The NHIS offers free nationwide health examination program to all members ≥ 40 years old every 2 years. The NHIS-HEALS database included 541,866 individuals, comprising a random selection of 10% of all health screening participants aged 40–79 years old in 2002 and 2003. It contains serial information about health examinations in 2002–2015 including physical examinations for blood pressure and body mass index (BMI) and questionnaires of medical history and lifestyle. It also includes individuals' health claim data about hospital visits, medical procedures, diagnosis, and drug prescriptions in 2002–2015. Any diagnosis made at each hospital visit was recorded according to the International Statistical Classification of Diseases and Related Health Problems 10th Revision (ICD-10). The NHIS-HEALS data were fully de-identified and anonymized for all analyses.

### Study patients

Based on the health claim codes in the NHIS-HEALS, we selected patients who underwent endovascular coil embolization (M1661, M1662) or surgical clipping (S4641, S4642) for

the treatment of cerebral aneurysm in 2002–2015 [9]. Among them, we only included patients with at least one available data of systolic and diastolic blood pressure (SBP and DBP, respectively) from the nationwide health examination program after the index date of cerebral aneurysm treatment (admission date for coil embolization or surgical clipping) and before the study end date (date of death, development of primary outcome, loss of eligibility for NHIS due to emigration, or Dec 31, 2015, whichever occurred the earliest). A flow chart of the patient inclusion process is shown in Fig 1. This study was approved, and informed consent was waived by the Institutional Review Board of Bundang CHA Medical Center (CHAMC 2018-05-011) due to the retrospective nature of the study and use of de-identified data.

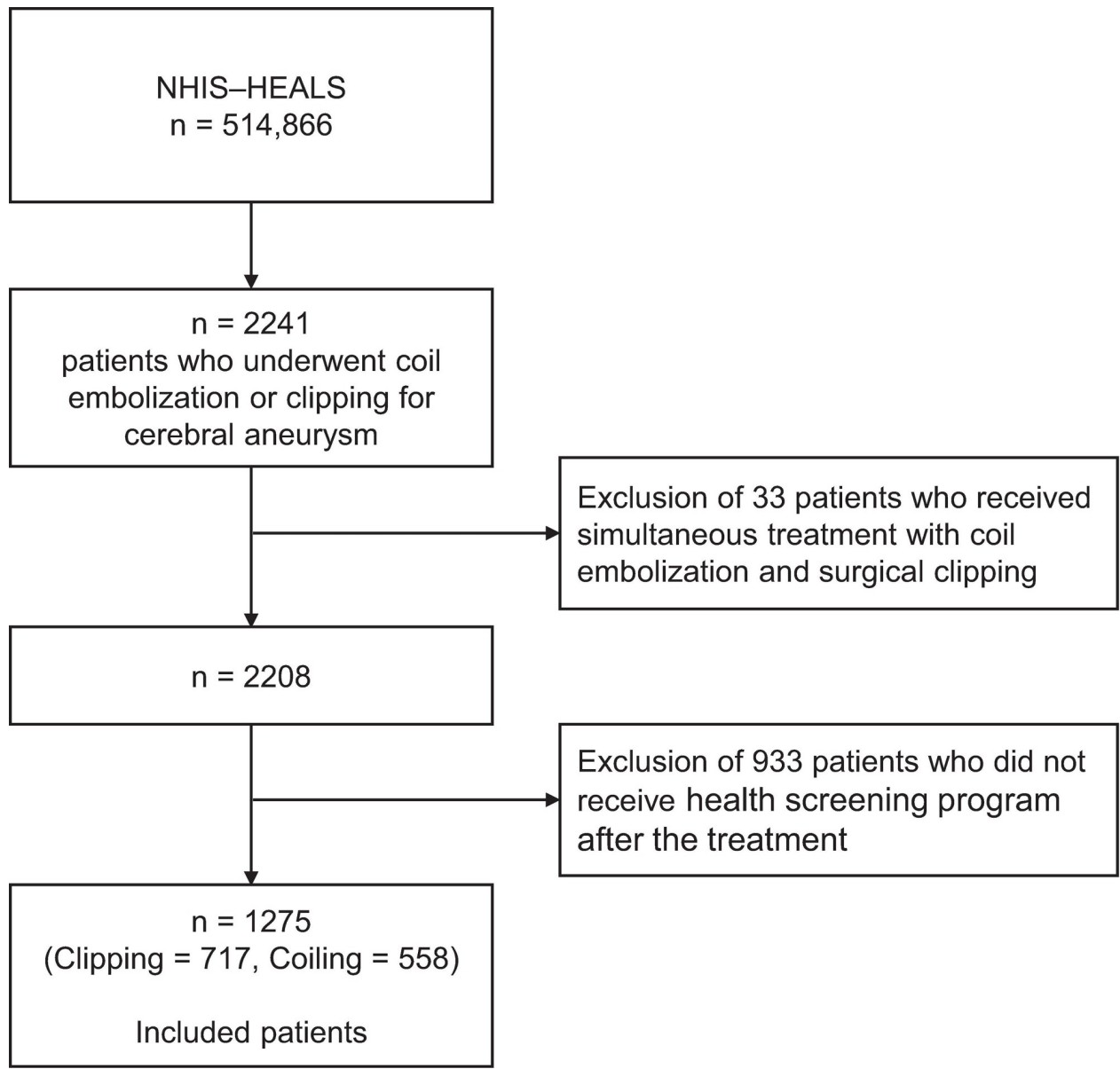

**Fig 1. Flow chart of patient inclusion process.** NHIS-HEALS, National Health Insurance Service-National Health Screening Cohort in Korea.

## Measure of blood pressure

Since the health screening program was recommended every 2 years, we were able to obtain longitudinal data of blood pressure of the study patients. According to the protocol, SBP and DBP were measured by qualified medical personnel at the brachial artery after a 5-minute rest in the sitting position using a cuff of appropriate size [10, 11]. Both automated oscillometric devices and mercury sphygmomanometers were used for blood pressure measurements. The choice of blood pressure measurement devices was left to the discretion of individual health examination centers. More information on blood pressure measurements during health check-ups in Korea is available in the reference [10]. In the analysis, we treated blood pressures as: 1) continuous variables (per 10 mmHg); and 2) categorical variables (SBP: ≤119, 120–129, 130–139, 140–149, ≥150 mmHg; DBP: ≤69, 70–79, 80–89, 90–99, ≥100 mmHg). In the categorical analysis, the most frequent categories of SBP (130–139 mmHg) and DBP (80–89 mmHg) were set as the references.

## Study outcomes

The primary outcome was defined as the composites of stroke, myocardial infarction (MI), and all-cause mortality, whichever occurred first. The development of stroke was defined as admission with a primary diagnosis of I60–I64 (hemorrhagic, I60–I62; ischemic, I63; not specified, I64) and the completion of brain computed tomography or magnetic resonance imaging during the admission period [9, 12, 13]. MI was defined as admission with a primary diagnosis of I21 [14]. The diagnostic accuracy of stroke and MI based on the health claim records in the NHIS was validated in prior studies [15, 16]. Each patient's date of death available in the NHIS-HEALS was linked with Statistics Korea.

## Covariates

Data on sex, treatment modality of cerebral aneurysm (coil embolization or surgical clipping), and age at treatment were collected. The treatments for ruptured and non-ruptured aneurysm were determined by the presence of the diagnostic code of nontraumatic SAH (I60) before or at the index date [9]. Based on the prescription claims data of individual patient, the use of antihypertensive medication was determined whether or not the patient was prescribed oral antihypertensive medications (any of calcium channel blocker, angiotensin-converting enzyme inhibitor, angiotensin receptor blocker, diuretics, and β-blocker) at the day of health examination. The self-administered questionnaire in the health examination included questions regarding lifestyle factors including current smoking status, alcohol consumption, and physical activity [8, 17]. If patients had a fasting blood glucose level ≥ 126 mg/dL or answered "yes" in the corresponding health screening questionnaire about the diagnosis of diabetes mellitus (DM), the patients were considered to have DM [18]. BMI was calculated by dividing weight by height squared as a continuous variable (kg/m$^2$). Along with SBP and DBP, the data obtained from the repeatedly health examination program were treated as time-dependent covariates.

## Statistical analysis

The collected data were expressed as number (%) for categorical variables and mean ± standard deviation for continuous variables. We performed time-dependent Cox proportional hazard regression analysis for the primary outcome. Due to the high correlation between SBP and DBP, two individual models with SBP and DBP were constructed. In the Cox regression model, repeatedly measured data at the health screening examination including blood pressure were treated as time-dependent variables. The proportional hazards

assumption for systolic/diastolic blood pressure in the Cox models was tested by calculating the Schoenfeld residuals, which was found to be satisfactory. In the multivariate models, adjustments were made for sex, age, treatment modality for cerebral aneurysm (coil embolization or surgical clipping), ruptured versus unruptured cerebral aneurysm, antihypertensive medication, the presence of DM, BMI, current smoking, physical activity, and alcohol consumption. We illustrated curves for the estimated risk with the continuous value of blood pressure using the restricted cubic spline method. In the secondary outcome analyses, we constructed cause-specific Cox regression models that had stroke, MI, and all-cause mortality as events of interest (each event serves as a competing risk). Data manipulation and statistical analyses were performed using SAS version 9.4 (SAS Institute, Cary, NC, USA) and R software version 3.3.3 (The R Foundation for Statistical Computing, Vienna, Austria; http://www.R-project.org/). Two-sided values of $p < 0.05$ was considered statistically significant.

## Results

### Clinical characteristics

According to the inclusion criteria (Fig 1), we finally included 1275 patients who underwent endovascular coil embolization (n = 558) and surgical clipping (n = 717). Among the included patients, 33.5% were male and the mean age at treatment was 59.4 ± 8.1 years. The patients' clinical characteristics are shown in Table 1.

### Distribution of blood pressure during follow-up

After treatment, mean follow-up duration was 6.13 ± 3.41 years. During the study period, there were a total of 3546 blood pressure measurements. The median number of blood pressure measurements per patient was 2 (interquartile range, 1–4). Among the included patients, the mean ± standard deviation time of the first and last health examination were 1.67 ± 1.57 years and 4.92 ± 3.35 years, respectively. We assessed blood pressure according to the time period since treatment for cerebral aneurysm. Fig 2 shows the distribution of blood pressure categories for each yearly blood pressure measurement (per 1-year interval).

Regardless of the time of measurement after the treatment for cerebral aneurysm, about 15–20% of patients had an SBP ≥ 140 mmHg and about 12–20% of patients had a DBP ≥ 90 mmHg among the patients who underwent blood pressure measurements at the time intervals. When we defined uncontrolled high blood pressure as either SBP ≥ 140 mmHg or DBP ≥ 90 mmHg, the proportion was 22.9% among the total 3546 times of blood pressure measurement in this study [19].

### Risk for primary outcome according to blood pressure

During the study period, 89 patients had the primary outcome (stroke, MI, all-cause death considering only the earliest event). In the univariate and multivariate Cox regression models, high SBP and DBP were significantly associated with increased risk of the primary outcome (Table 2). When blood pressures were treated as continuous variables, adjusted hazard ratio (HR) [95% confidence interval (CI)] for primary outcome per 10 mmHg increase in SBP and DBP were 1.16 [1.01–1.35] and 1.32 [1.06–1.64], respectively. There was no significant interaction effect between blood pressure and antihypertensive medication on the risk for primary outcome (p-value for interaction between systolic blood pressure and antihypertensive medication = 0.477, p-value for interaction between diastolic blood pressure and antihypertensive medication = 0.808).

Fig 3 illustrates a spline curve for HR according to blood pressure. In the spline curves, risk of the primary outcome seems to increase with SBP > 130 mmHg and DBP > 80 mmHg.

**Table 1. Clinical characteristics of included patients.**

| Variable | All patients | Ruptured | Non-ruptured | p-value† |
|---|---|---|---|---|
| Number of included patients | 1275 | 551 | 724 | - |
| Total number of health examinations | 3546 | 1825 | 1721 | - |
| *Variables at treatment of cerebral aneurysm* | | | | |
| Sex, male | 427 (33.5) | 193 (35.0) | 234 (32.3) | <0.001 |
| Age, years | 59.4 ± 8.1 | 57.0 ± 8.1 | 61.3 ± 7.6 | <0.001 |
| Treatment modality | | | | <0.001 |
| Surgical clipping | 717 (56.2) | 378 (68.6) | 339 (46.8) | |
| Coil embolization | 558 (43.8) | 173 (31.4) | 385 (53.2) | |
| Cerebral aneurysm | | | | - |
| Ruptured | 551 (43.2) | 551 (100) | - | |
| Non-ruptured | 724 (56.8) | - | 724 (100) | |
| *Variables at first health examination after cerebral aneurysm treatment* * | | | | |
| Systolic blood pressure, mmHg | 127.1 ± 14.7 | 128.9 ± 14.6 | 125.7 ± 14.6 | <0.001 |
| Diastolic blood pressure, mmHg | 78.5 ± 9.6 | 79.6 ± 9.9 | 77.6 ± 9.3 | <0.001 |
| Antihypertensive medication | 653 (51.2) | 268 (48.6) | 385 (53.2) | <0.121 |
| Diabetes mellitus | 262 (20.6) | 92 (16.7) | 170 (23.5) | 0.004 |
| Body mass index (kg/m$^2$) | 24.2 ± 2.9 | 24.1 ± 2.9 | 24.2 ± 2.9 | 0.551 |
| Current smoking | 108 (8.5) | 50 (9.1) | 58 (8.0) | 0.566 |
| Alcohol consumption, frequency per week | | | | <0.001 |
| <1 time | 787 (61.7) | 382 (69.3) | 405 (55.9) | |
| 1–2 times | 403 (31.6) | 142 (25.8) | 261 (36.0) | |
| 3–4 times | 63 (4.9) | 19 (3.4) | 44 (6.1) | |
| ≥5 times | 22 (1.7) | 8 (1.5) | 14 (1.9) | |
| Exercise, days per week | | | | 0.257 |
| <1 day | 648 (50.8) | 291 (52.8) | 357 (49.3) | |
| 1–3 days | 365 (28.6) | 158 (28.7) | 207 (28.6) | |
| ≥5 days | 262 (20.6) | 102 (18.5) | 160 (22.1) | |

Data are represented as number of patients (%) or mean ± standard deviation.

*Treated as time-dependent covariates in the survival analysis.

†p-value are derived by chi-square test or independent t-test between groups with ruptured and non-ruptured aneurysm.

In the analysis of blood pressure categories (Table 2), groups with high blood pressure were positively associated with risk of the primary outcome. Compared to SBP of 130–139 mmHg (reference), SBP of 140–149 mmHg (adjusted HR, 2.41; 95% CI, 1.18–4.94) and SBP ≥ 150 mmHg (adjusted HR, 3.78; 95% CI, 1.93–7.42) were at significantly increased risk of the primary outcome. Interestingly, there was an increased risk at an SBP ≤ 119 mmHg (adjusted HR, 2.04; 95% CI, 1.09–3.83) compared to an SBP of 130–139 mmHg, suggesting a J-shaped curve relationship. In the analysis of the DBP category, the risk of primary outcome increased in proportion to the DBP. Compared to the reference (DBP 80–89 mmHg), adjusted HR [95% CI] for DBP of 90–99 mmHg and ≥100 mmHg were 2.68 [1.47–4.89] and 2.92 [1.28–6.69], respectively. For DBP, the J-shaped curve relationship was not significant.

## Secondary outcome analysis

In the secondary outcome analysis for individual outcomes (Table 3), both high SBP and DBP were significantly associated with increased risk of all stroke (I60–I64) after cerebral aneurysm treatment. A high DBP was an independent risk factor for hemorrhagic stroke (I60–I62;

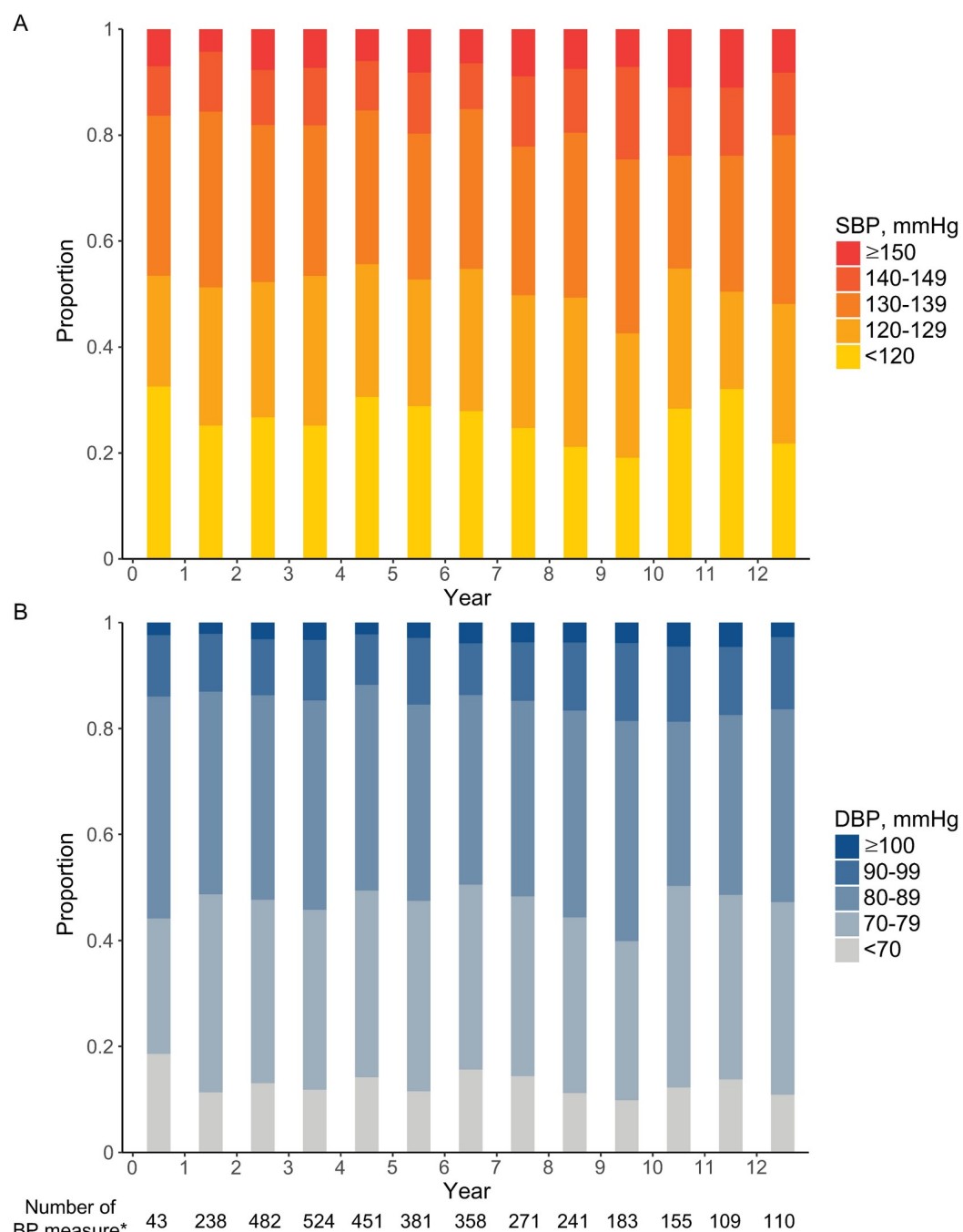

**Fig 2. Distribution of blood pressure according to the time from the treatment of cerebral aneurysm.** Proportion of category by (A) systolic blood pressure and (B) diastolic blood pressure among patients who underwent a health examination at each time period since the treatment for cerebral aneurysm. The bar between years 1 and 2 (x-axis) indicates the distribution of blood pressure categories among the patients who underwent a health examination at 1–2 years after treatment for cerebral aneurysm. *Number of blood pressure measurements during the time period. BP, blood pressure (mmHg); DBP, diastolic blood pressure; SBP, systolic blood pressure.

adjusted HR, 1.56; 95% CI, 1.11–2.20). The risk of hemorrhagic stroke was also positively (but not significantly) associated with high SBP (adjusted HR, 1.21; 95% CI, 0.97–1.52).

**Table 2. Effects of blood pressure on primary outcome after aneurysm treatment.**

|  | Crude HR [95% CI] | Adjusted HR [95% CI] |
|---|---|---|
| SBP *as continuous variable*, per 10 mmHg | 1.19 [1.04–1.36] | 1.16 [1.01–1.35] |
| SBP *as categorical variable*, mmHg |  |  |
| ≤119 | 1.95 [1.05–3.60] | 2.04 [1.09–3.83] |
| 120–129 | 0.97 [0.47–2.01] | 1.00 [0.48–2.12] |
| 130–139 | Ref | Ref |
| 140–149 | 2.49 [1.23–5.05] | 2.41 [1.18–4.94] |
| ≥150 | 3.86 [1.95–7.64] | 3.78 [1.93–7.42] |
| DBP *as continuous variable*, per 10 mmHg | 1.30 [1.06–1.59] | 1.32 [1.06–1.64] |
| DBP *as categorical variable*, mmHg |  |  |
| ≤69 | 1.23 [0.59–2.56] | 1.18 [0.57–2.46] |
| 70–79 | 1.27 [0.74–2.17] | 1.28 [0.75–2.17] |
| 80–89 | Ref | Ref |
| 90–99 | 2.39 [1.30–4.39] | 2.68 [1.47–4.89] |
| ≥100 | 2.87 [1.24–6.64] | 2.92 [1.28–6.69] |

Data are derived from time-dependent Cox proportional hazard regression models. Primary outcome is a composite of stroke, myocardial infarction, and all-cause death. Adjustment was done for sex, age, coil embolization/surgical clipping, ruptured/unruptured cerebral aneurysm, the presence of diabetes mellitus, antihypertensive medication, body mass index, current smoking, physical activity, and alcohol consumption. **Abbreviations:** CI, confidence interval; DBP, diastolic blood pressure; HR, hazard ratio; SBP, systolic blood pressure.

## Sensitivity analysis

To investigate the potential interaction effects between high blood pressure and other risk factors, we performed a subgroup analysis according to the presence of individual risk factors (Fig 4). In the subgroup analysis, we found no significant interaction between SBP/DBP and any other risk factors.

## Comparison of included and excluded patients according to the available data of health examination after the treatment

In current study, we only included patients who underwent at least one health examination program. To evaluate the difference of prognosis in the included and excluded patients according to availability of health examination data (Fig 1), we compared the primary outcome between the groups (Table 4). In the comparison, there was significantly increased risk for primary outcome in the excluded patients who did not receive health examination after the treatment than included patients.

## Discussion

The current study evaluated real-world data of longitudinal blood pressure after the treatment of cerebral aneurysm and the relationship with clinical outcome using data from the nationwide health screening database. We found that high blood pressure was prevalent even in patients with cerebral aneurysm who underwent coil embolization or clipping. Our finding is in line with those of prior epidemiologic studies showing that blood pressure control is frequently insufficient in clinical practice regardless of general or disease-specific populations [20–22]. The prevalent feature of high blood pressure showed clear adverse effects on long-

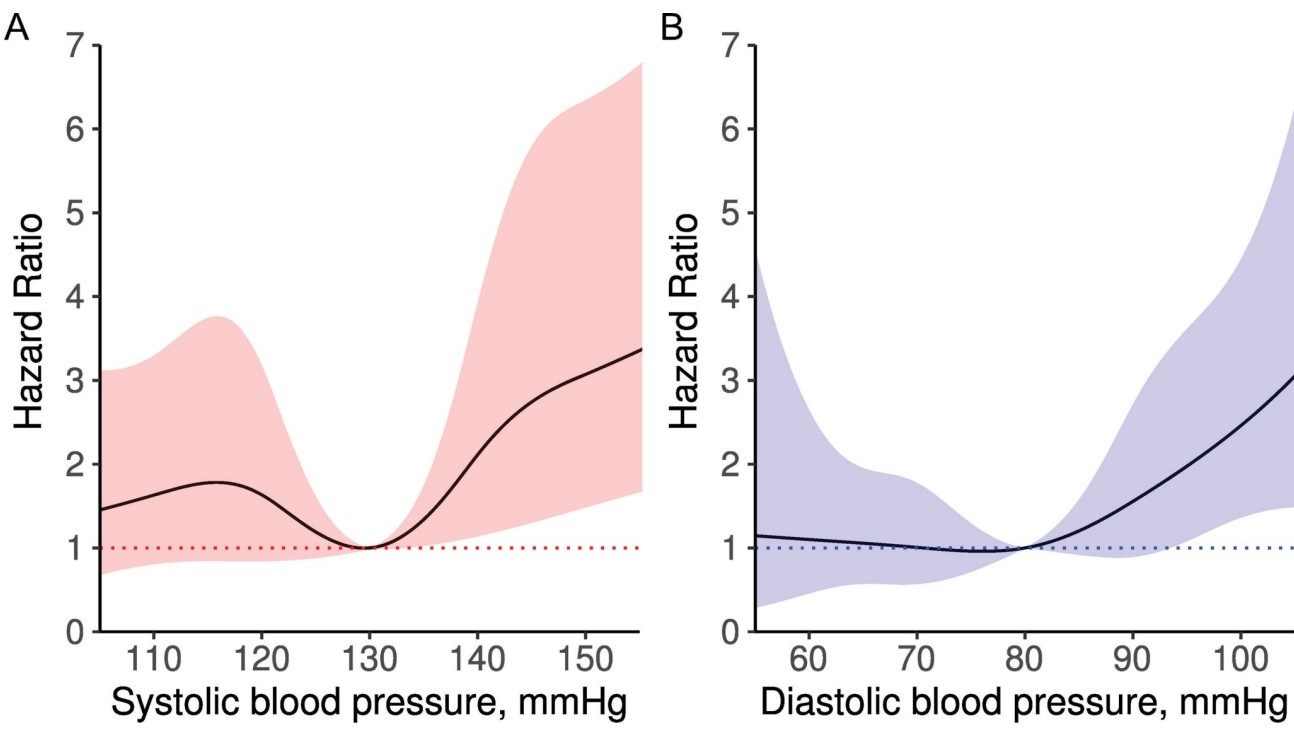

**Fig 3. Spline curves of the association between blood pressure and risk of primary outcome.** Hazard ratio and 95% confidence interval for (A) SBP and (B) DBP are presented with solid lines and shaded areas. The data are derived from time-dependent Cox proportional hazard regression model for primary outcome using the method of restricted cubic splines with knots at 110, 120, 130, 140, and 150 mmHg of SBP and at 60, 70, 80, 90, and 100 mmHg of DBP. The reference is SBP 130 mmHg and DBP 80 mmHg. DBP, diastolic blood pressure; SBP, systolic blood pressure.

term prognosis. These findings suggest that more attention should be paid to achieving optimal longitudinal blood pressure control after treatment for cerebral aneurysm.

Inadequate blood pressure control is a multifactorial problem that includes unawareness of hypertension, inaccurate measurements, nonadherence to antihypertensive medications, suboptimal therapeutic goal creation, low socioeconomic status, environmental factors, health care system issues, unhealthy lifestyles, excessive salt consumption, cultural and behavioral influences, and other factors [23]. For better control of blood pressure, a multilevel systemic

**Table 3. Secondary outcome analysis of individual outcomes by blood pressure after aneurysm treatment.**

| Secondary outcome (number of events) | Adjusted HR [95% CI] per 10 mmHg of SBP | Adjusted HR [95% CI] per 10 mmHg of DBP |
|---|---|---|
| All stroke (n = 57)* | 1.22 [1.02–1.47] | 1.40 [1.07–1.84] |
| Hemorrhagic stroke (n = 33) | 1.21 [0.97–1.52] | 1.56 [1.11–2.20] |
| Ischemic stroke (n = 21) | 1.29 [0.94–1.76] | 1.24 [0.76–2.01] |
| Myocardial infarction (n = 6) | 1.16 [0.78–1.73] | 0.72 [0.36–1.43] |
| All-cause death (n = 26) | 1.06 [0.79–1.41] | 1.31 [0.88–1.97] |

Models are adjusted for sex, age, coil embolization/surgical clipping, ruptured/unruptured cerebral aneurysm, the presence of diabetes mellitus, antihypertensive medication, body mass index, current smoking, physical activity, and alcohol consumption.

*All stroke is defined as hemorrhagic stroke (I60–I62), ischemic stroke (I63), and stroke not specified as hemorrhage or infarction (I64) according to the International Classification of Diseases 10[th] Revision. **Abbreviations:** CI, confidence interval; HR, hazard ratio.

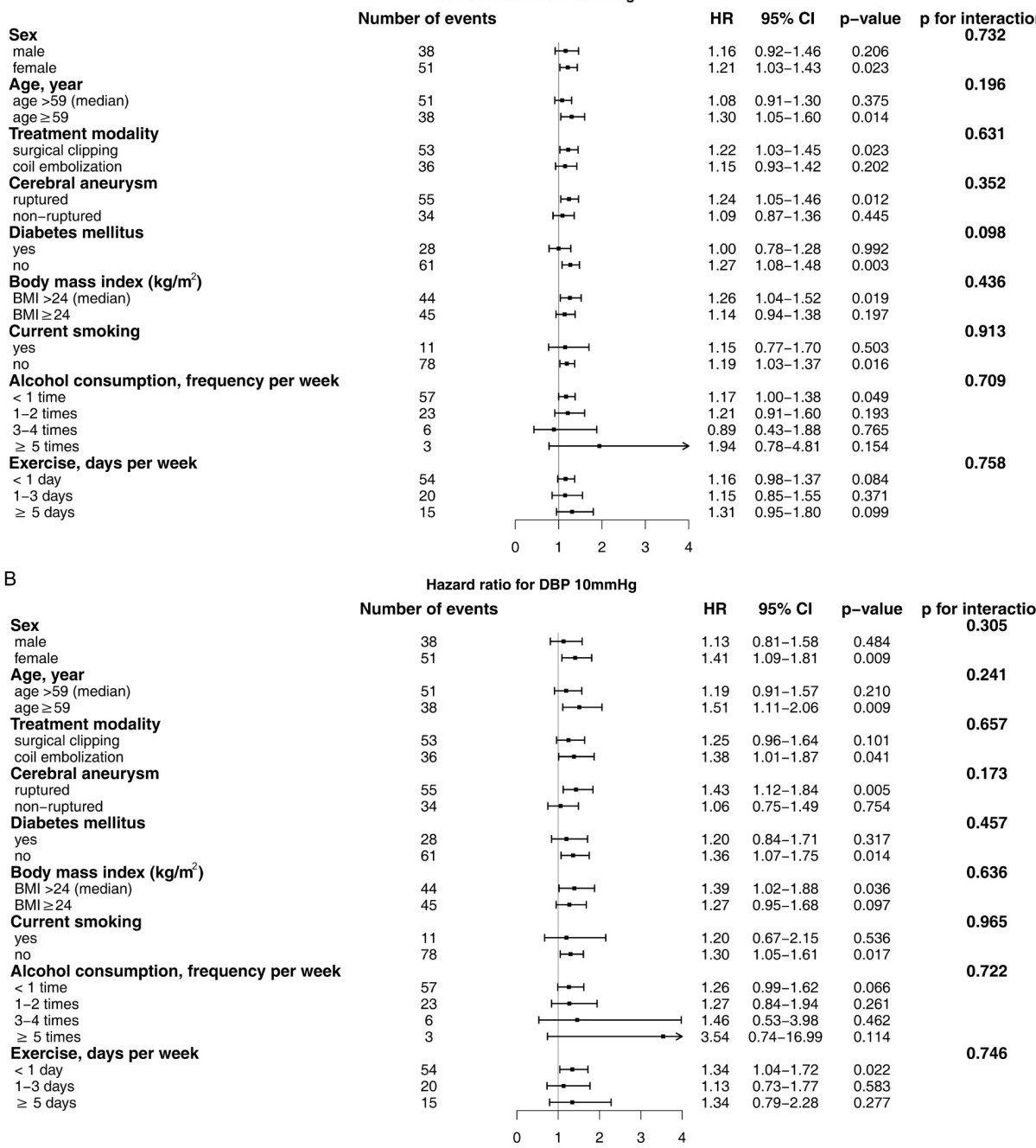

**Fig 4. Effect of blood pressure on primary outcome by subgroup.** Data are HR and 95% CI for (A) SBP and (B) DBP derived from time-dependent Cox proportional hazard regression model for primary outcome. *p-value for interaction between risk factor and blood pressure. CI, confidence interval; DBP, diastolic blood pressure; HR, hazard ratio; SBP, systolic blood pressure.

approach is recommended, such as enhancing awareness and monitoring blood pressure, implementing healthy lifestyle choices, and improving motivation and adherence to antihypertensive medications [24]. We previously reported that poor adherence to antihypertensive medication was significantly associated with poor prognosis of patients treated for cerebral aneurysm and those with SAH [9, 12].

**Table 4. Comparison of the event outcome between included and excluded patients according to availability of health examination data after treatment to cerebral aneurysm.**

| Group | Number of patients | Patient-years at risk | Number of events* | p-value† |
|---|---|---|---|---|
| Included patients (study participants) | 1275 | 7816 | 89 | <0.001 |
| Excluded patients due to no available data of health screening | 933 | 989 | 589 | |

*composite of stroke, myocardial infarction, and all-cause death.

† p-value derived by log-rank test.

There are conflicting epidemiological reports of the relationship between hypertension and cerebral aneurysm, but a large proportion of data support the important role of high blood pressure on the development and prognosis of cerebral aneurysm [3, 5, 25]. In a case-control study of 20,767 Medicare patients, the prevalence of hypertension was significantly higher in patients with unruptured cerebral aneurysms compared to a control population (43.2% vs 34.4%, respectively) [26]. Hypertension was more prevalent in those with multiple aneurysms [27]. In a longitudinal follow-up computed tomography angiography study of 610 patients who survived after SAH, the presence of hypertension was a significant risk factor for new aneurysm formation and enlargement of the known aneurysm [28]. In the large population-based prospective cohort study of 74,977 participants (Nord-Trøndelag Health study), there was a strong positive association between high blood pressure (SBP and DBP) and risk of aneurysmal SAH [6]. In the UIATS (unruptured intracranial aneurysm treatment score) model developed as a decision guideline for the appropriate management of unruptured cerebral aneurysm, hypertension is a key factor favoring aneurysm repair due to a high risk of rupture rather than conservative management [29]. Based on these epidemiologic evidence, hypertension is recognized as an important modifiable risk factor for preventing cerebral aneurysm rupture [5]. Furthermore, the presence of hypertension before SAH is a risk factor for rebleeding and worse clinical outcome in the SAH patients [30]. Current guidelines for cerebral aneurysm strongly recommend blood pressure monitoring to patients with cerebral aneurysm, and those with high blood pressure should receive treatment for hypertension [5, 31, 32].

There are suggested mechanisms regarding how hypertension can play an important role in aneurysmal SAH [32]. Exposure to chronic hypertension causes intimal thickening, necrosis of the tunica media, vascular inflammation, changes in the compositional matrix, and degeneration of the elastic lamina of the arterial wall, which lead to focal weakening and dilatation of cerebral artery wall [33]. The increase in systemic blood pressure can directly lead to a simultaneous increase in intra-aneurysm blood pressure [34]. The normalization of blood pressure in a mice model with aneurysm formation significantly reduced the risk of aneurysmal rupture [7]. Beyond mechanical stress, activation of the local renin-angiotensin system and toll-like receptor 4 pathway by systemic hypertension can induce vascular inflammation and remodeling, contributing to aneurysm rupture [35]. Increased activity and imbalances of the metalloproteinase/tissue inhibitors of metalloproteinase system in hypertensive patients predisposes them to cerebral aneurysm formation and growth [36, 37]. Hypertension also contributes to disturbances in the brain nitric oxide synthase system, which participates in oxidative stress, arterial damage, remodeling, and cerebral aneurysm formation and rupture [38].

The optimal blood pressure target among patients treated with cerebral aneurysm is unknown. The updated American College of Cardiology/American Heart Association (ACC/AHA) guidelines recommend strict blood pressure control (target of SBP < 130 mmHg and DBP < 80 mmHg) in patients considered at high cardiovascular risk [39]. The 2018 European

Society of Cardiology and European Society of Hypertension (ESC/ESH) guidelines on the management of hypertension also recommended that the treated blood pressure values should be targeted to 130/80mmHg or lower in most patients [40]. Unfortunately, there is no specific comment about the target blood pressure of patients with cerebral aneurysm in the ACC/AHA and ESC/ESH guidelines. Our current study demonstrates that high SBP ($\geq$ 140 mmHg) and DBP ($\geq$ 90 mmHg) significantly increase risk for primary outcomes compared to the reference group (SBP 130–139 mmHg, DBP 80–89 mmHg). Therefore, we supposed that the target pressure should be at least SBP < 140 mmHg and DBP < 90 mmHg considering the significantly increased risk over those levels. Regarding the lowest risk around SBP of 130 mmHg and DBP of 80 mmHg in our data, we supposed that the target of SBP of 130 mmHg and DBP of 80 mmHg recommended in the recent ACC/AHA and ESC/ESH hypertension guidelines for those at high cardiovascular risk was reasonable for patients treated with cerebral aneurysm.

Interestingly, we found an increased risk in the category of SBP < 120 mmHg versus reference (J-curve phenomenon). The J-curve phenomenon is a controversial issue, but the phenomenon have been reported in both observational and randomized trials among the general population and patients with vascular disease [41–43]. The exact mechanism of the J-curve risk pattern is not well known. As one suggested mechanism, some deteriorating conditions (malignancy, infection, malnutrition, heart failure, etc.) are more frequently found in patients with low blood pressure [44]. A predisposition to falling, decreased perfusion to vital organs, or adverse events related to hypotension may increase an individual's cardiovascular risk and mortality [45]. The 2018 ESC/ESH hypertension guidelines against SBP targeted to < 120 mmHg in high-risk patients considering balance of benefit versus harm [40]. Although strict blood pressure control is recommended for patients with cerebral aneurysm, considerable caution should be used to avoid hypotension.

## Strengths and limitations

The present study has strengths and limitations. Because the NHIS is a single payer insurer in Korea, we could include patients who underwent coil embolization or surgical clipping for cerebral aneurysm in a nationwide population and evaluate the long-term prognosis of them. The nationwide health screening examination is performed repeatedly, which enables the collection of longitudinal data on blood pressure. Beyond the strengths, we should acknowledge the several limitations of this study. First, given the limitation of retrospective observational cohort, our study could not prove the efficacy of interventions of improving blood pressure control in patients with cerebral aneurysm. Due to the lack of detailed clinical information from the health screening database, we could not assess cerebral aneurysm characteristics (size, location, morphology). In current study, there was a large number of excluded patients who never attended a health examination program after treatment for cerebral aneurysm. Early mortality or severe disability after treatment of cerebral aneurysm can limit participation to the health examination program. Indeed, the excluded patients had significantly worse prognosis than included patients. Therefore, it should be noted that the included patients may have a selection bias and did not adequately represent the population who underwent treatment to cerebral aneurysm. Despite the long-term follow-up, only a small number of patients experienced clinical outcomes, which limited statistical power and further analysis of individual outcomes and interaction between covariates. The current study was based on patients who underwent coil embolization or surgical clipping. The general application of our result to all patients with cerebral aneurysm should be further evaluated. Although we assessed clinical outcomes using the definitions validated in prior studies, the accuracy based on health claims codes might be limited. Our study population was Korean patients. There might be differences

in decision making to treat aneurysm and disease susceptibility with different genetic and ethnic populations. Therefore, our data must be interpreted with caution.

## Conclusions

For patients with treated cerebral aneurysm, high SBP ($\geq$ 140 mmHg) and DBP ($\geq$ 90 mmHg) were prevalent and significantly associated with worse clinical outcome. Regular monitoring and strict control of blood pressure may improve the prognosis of the patients with treated cerebral aneurysm.

## Acknowledgments

This study used the dataset of NHIS-HEALS (NHIS-2018-2-236) created by the National Health Insurance Sharing Service.

## Author Contributions

**Conceptualization:** Jinkwon Kim, Jang Hoon Kim, Sang Hyun Suh, Kyung-Yul Lee.

**Data curation:** Jinkwon Kim, Hye Sun Lee.

**Formal analysis:** Jinkwon Kim, Hye Sun Lee.

**Funding acquisition:** Jinkwon Kim.

**Investigation:** Sang Hyun Suh, Kyung-Yul Lee.

**Methodology:** Jinkwon Kim, Jang Hoon Kim, Hye Sun Lee, Sang Hyun Suh.

**Project administration:** Sang Hyun Suh, Kyung-Yul Lee.

**Resources:** Kyung-Yul Lee.

**Software:** Kyung-Yul Lee.

**Supervision:** Jang Hoon Kim, Hye Sun Lee, Sang Hyun Suh, Kyung-Yul Lee.

**Validation:** Hye Sun Lee.

**Visualization:** Jinkwon Kim, Hye Sun Lee.

**Writing – original draft:** Jinkwon Kim, Jang Hoon Kim.

**Writing – review & editing:** Sang Hyun Suh, Kyung-Yul Lee.

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
