## [Decision Letter · Decision Letter 0]

9 Dec 2020

PONE-D-20-33566

Association between longitudinal blood pressure and prognosis after treatment of cerebral aneurysm: A nationwide population-based cohort study

PLOS ONE

Dear Dr. Lee,

Thank you for submitting your manuscript to PLOS ONE. After careful consideration, we feel that it has merit but does not fully meet PLOS ONE’s publication criteria as it currently stands. Therefore, we invite you to submit a revised version of the manuscript that addresses the points raised during the review process.

Specifically, you might have to address the comments of the two Reviewers, especially on the concern of the outcome of the excluded subjects. Please, supply information on the device type for the blood pressure measurement, and on the number of events and patients in the subgroups showing in the figure 4.

We look forward to receiving your revised manuscript.

Kind regards,

Yan Li, MD, PhD

Academic Editor

PLOS ONE

Journal Requirements:

Reviewers' comments:

Reviewer's Responses to Questions

**Comments to the Author**

1. Is the manuscript technically sound, and do the data support the conclusions?

Reviewer #1: Yes

Reviewer #2: Yes

2. Has the statistical analysis been performed appropriately and rigorously? 

Reviewer #1: Yes

Reviewer #2: Yes

3. Have the authors made all data underlying the findings in their manuscript fully available?

Reviewer #1: Yes

Reviewer #2: Yes

4. Is the manuscript presented in an intelligible fashion and written in standard English?

Reviewer #1: Yes

Reviewer #2: Yes

5. Review Comments to the Author

Reviewer #1: 1. Of 2241 patients who underwent coil embolization or clipping for cerebral aneurysm in 2002-2015, 966 (43.1%) patients were excluded mostly because of absent health screening after treatment. This segment of the population may accept different medical care and get different outcome. And the included patients perhaps can't represent the

target population.

2. Time-dependent Cox proportional hazard regression analysis for the primary outcome were performed with SBP or DBP from 3546 times of BP measurement in 1275 included patients during 1-13 years. It was unknown the data distribution information of the measurement time. The HR can be considered as a weighted average of short-term effects on mortality(outcomes). Basically, in a time-dependent analysis, the follow-up time for each patient is divided into different time windows. For each time –window, a separate Cox analysis is carried out using the specific value of the time-dependent variable at the beginning of that specific time window. Then a weighted average of all the time window-specific results is calculated. This weighted average of a series of relatively short-term effects is presented as the result of the analysis as one RR. In this paper, , the time of BP management in different time windows were varied and uncertain, which would decrease the value of HR.

Reviewer #2: The authors evaluated the association of blood pressure and adverse outcome in 1275 cerebral aneurysm patients who had participated in health screening after interventional procedure. With a mean follow-up of 6.13 years and a median repeated blood pressure reading of 2 times, high blood pressure was moderately associated with poor composite outcomes of stoke, myocardial infarction and all-cause death.

Overall the paper provides additional insights to blood pressure management of cerebral aneurysmal patients. That said, some comments must be addressed.

1. As Figure 1 illustrates, 933 out of 2241 patients were screened but excluded due to non-attendance of health screening. Given short-term mortality and disability are high in SAH patients, the composite outcome in the excluded patients should be demonstrated.

2. As per Table 1, 43.2% of the included had ruptured aneurysm, who should possibly have "SAH a life-threatening stroke" before the treatment procedure and were at high risk of future adverse outcome. Please provide number of event at each subgroups of Figure 4, and provide detailed comparison in Table 1 as grouped by the status of cerebral aneurysm rupture.

3. Antihypertensive medication could not be overlooked given the limited number of blood pressure measurement. In line 73-74 the authors wrote, "It also includes individuals’ health claim data about hospital visits, medical procedures, diagnosis, and drug prescriptions in 2002–2015." The authors should take antihypertensive medication into account if relevant data are available.

6. PLOS authors have the option to publish the peer review history of their article (what does this mean?). If published, this will include your full peer review and any attached files.

Reviewer #1: No

Reviewer #2: No

---

## [Author Response · Author response to Decision Letter 0]

14 Jan 2021

Response to the reviewer comments is included in an attached word file.

Sincerely.

---

## [Decision Letter · Decision Letter 1]

10 May 2021

Association between longitudinal blood pressure and prognosis after treatment of cerebral aneurysm: A nationwide population-based cohort study

PONE-D-20-33566R1

Dear Dr. Lee,

We’re pleased to inform you that your manuscript has been judged scientifically suitable for publication and will be formally accepted for publication once it meets all outstanding technical requirements.

Kind regards,

Yan Li, MD, PhD

Academic Editor

PLOS ONE

Additional Editor Comments (optional):

Please provide more information about the methods of blood pressure measurement in this study. Which kind of devices were used, oscillometric or mercury sphygmomenometer?

Reviewers' comments:

Reviewer's Responses to Questions

**Comments to the Author**

1. If the authors have adequately addressed your comments raised in a previous round of review and you feel that this manuscript is now acceptable for publication, you may indicate that here to bypass the “Comments to the Author” section, enter your conflict of interest statement in the “Confidential to Editor” section, and submit your "Accept" recommendation.

Reviewer #2: (No Response)

2. Is the manuscript technically sound, and do the data support the conclusions?

Reviewer #2: Yes

3. Has the statistical analysis been performed appropriately and rigorously? 

Reviewer #2: Yes

4. Have the authors made all data underlying the findings in their manuscript fully available?

Reviewer #2: Yes

5. Is the manuscript presented in an intelligible fashion and written in standard English?

Reviewer #2: Yes

6. Review Comments to the Author

Reviewer #2: Please include p values in Table 2 and Table 3 and in the manuscript where relevant values were referred to.

7. PLOS authors have the option to publish the peer review history of their article (what does this mean?). If published, this will include your full peer review and any attached files.

Reviewer #2: No

---

## [Editor Report · Acceptance letter]

19 May 2021

PONE-D-20-33566R1 

Association between longitudinal blood pressure and prognosis after treatment of cerebral aneurysm: A nationwide population-based cohort study 

Dear Dr. Lee:

I'm pleased to inform you that your manuscript has been deemed suitable for publication in PLOS ONE. Congratulations! Your manuscript is now with our production department. 

Kind regards, 

on behalf of

Professor Yan Li 

Academic Editor

PLOS ONE